# Identifying and Correcting Signal Shift in DMSP-OLS Data

**Konstantin Ash** [1],*  **and Kevin Mazur** [2]

[1] School of Politics, Security and International Affairs, University of Central Florida, Orlando, FL 32816, USA
[2] Department of Political Science, Northwestern University, Evanston, IL 60208, USA;
kevin.mazur@northwestern.edu
\* Correspondence: konstantin.ash@ucf.edu; Tel.: +1-407-823-2608

**Abstract:** Defense Meteorological Satellite Program Operational Linescan System (DMSP-OLS) nighttime light data has become a key tool of the environmental and social scientific fields, but suffers from several validity problems. We highlight one such problem—shifts in the digital number position in DMSP-OLS composites in the same satellite. We present techniques for identifying the problem, using moving window raster correlation and visual inspection, and for solving the problem, by assigning control points and manually shifting raster positions. To illustrate the importance of accounting for signal shift, we re-examine a recent analysis of the relationship between public goods provision and patterns of violence in the 2011 Syrian uprising and ensuing civil war. We find the statistical results change considerably when correcting for signal shift. We attribute this change to the systematic undercounting of light intensity in heavily populated areas. We close by identifying the types of research that would most benefit from our correction and suggest future refinements to our technique through automation.

**Keywords:** DMSP-OLS; signal shift; nighttime lights

## 1. Introduction

In the mid-1990s, the United States' National Oceanic and Atmospheric Association's (NOAA) Defense Meteorological Satellite Program Operational Linescan System (DMSP-OLS) began publicly releasing annual composites of nighttime lights, quantifying the relative intensity of light emissions on a global scale. DMSP-OLS nighttime lights data have been made available for each year between 1992 and 2013 [1]. Composites are created from daily images of outdoor lights, fires, and gas flares between 20:00 and 21:30 local time, dropping images obstructed by clouds or solar glare. Light intensity is measured by digital numbers (DNs) ranging from zero to 63 [2]. Researchers from numerous scientific disciplines have made use of these data as a proxy for a wide range of concepts, including population density [3–5], urbanization [6,7], electrification [8,9], political favoritism [10–12], environmental degradation [13], carbon dioxide emissions [14,15], energy consumption [16], and economic activity [17–19].

DMSP-OLS nighttime lights data also have several important limitations. They cannot capture variation in high-density urban areas due to pixel saturation [20] (though several corrections for saturation issues have been identified, including incorporating ground cover and census data [21]) and conflate electricity production with gas flares from oil production [22]. Comparisons of annual composites across time are also problematic due to signal decay and satellite changes [23]. While saturation and signal decay are no longer issues with the release of monthly images from the Visible Infrared Imaging Radiometer Suite (VIIRS) satellite [24], DMSP-OLS data remain of interest to researchers using historical data (e.g., [25,26]). As such, it is important to correct issues that may bias inferences for practitioners still using DMSP-OLS data.

Our study identifies an additional limitation in DMSP-OLS data: a systematic shift in the geospatial position of DNs throughout the lifetimes of individual satellites [27,28]. The discrepancy has already been observed by Tuttle et al. [27], who detected a nearly 3-km shift between the physical coordinates of where a source of light (placed by the researchers) was located and where it appears on nighttime light maps. Tuttle et al. [27] could not identify a technical cause for this shift. We extend this finding, identifying systematic differences in the positions of individual DNs in DMSP-OLS maps of Syria for the years 2007 and 2009.

Before continuing, we would like to address the reason why existing solutions for signal decay cannot be applied to signal shift. Signal decay is a decline in the effectiveness of satellite sensors over time [29], which results in fairly minor differences in DN resolution across years. It can be corrected through intercalibration—regression-based global algorithms that adjust value incompatibility across years and satellites (see [1,22,30]). Signal shift is not accounted for through intercalibration, which only adjusts the relative intensity of DNs, not their positions (see Figure S5).

We focus on Syria because it is the basis for an influential recent study by De Juan and Bank [31] examining the relationship between public goods provision and violence in the 2011 Syrian uprising. The study utilizes the difference between average DMSP-OLS nighttime lights in 2007 and 2009 in a given Syrian administrative region as a proxy for "load shedding"—power outages imposed by electricity providers to deal with grid voltage stability issues during a supply crisis [32]. This suggests that areas with less load shedding were favored by the regime and thus less likely to experience violence early in the uprising. The finding relies on the difference in nighttime light intensity from one year to another and could have been particularly affected by signal shift.

The study caught our attention because its results showed some of the highest drops in nighttime lights, and therefore the most load shedding, in Syria's Alawi heartland, including al-Qardaha, the hometown of the ruling al-Asad family. This finding runs contrary to both qualitative studies of patronage in Syria (e.g., [33–35]) and recent cross-national work on higher nighttime light intensity in leaders' hometowns [10].

For the remainder of our study, we outline how signal shift can be identified across two different annual DMSP-OLS maps, how to resolve this issue and how this problem can affect findings. In re-examining DMSP-OLS maps of Syria from 2007 and 2009, we find a digital number position shift of approximately 17 km to the south and 4.5 km to the east. Accounting for this shift improves the analysis of De Juan and Bank [31] by correcting for the systematic underestimation of light emissions in districts with higher populations.

## 2. Materials and Methods

Our first step in identifying signal shift is to obtain DMSP-OLS composites for the F16 satellite, which ranges from 2004 to 2009, from NOAA [36]. We employ a moving window technique [37,38] to obtain correlations for $5 \times 5$ pixel grid squares around every cell in each F16 raster composite pairing. This grid size accounts for the spatial correlation of the OLS, which has an effective isotropic field-of-view of, at best, 2.2 km [39], rendering at least every $4 \times 4$—and on average every $5 \times 5$—quadrant spatially correlated. The results in Table 1 indicate that the 2009 composite is markedly less correlated with other years from the same satellite. Similar correlation analysis on maps from the other five DMSP-OLS satellites (F10, F12, F14, F15, and F18) is shown on Tables S1–S5. For F12, F14 and F15, there is a pattern of the first and last years covered by a satellite being less correlated to the others.

**Table 1.** F16 Satellite Interannual Pearson Correlations.

| Year: | 2004 | 2005 | 2006 | 2007 | 2008 | 2009 (Unshifted) | 2009 (Shifted) |
|-------|------|------|------|------|------|------------------|----------------|
| 2004 | X | 0.573 | 0.585 | 0.630 | 0.608 | 0.558 | **0.500** |
| 2005 | | X | 0.659 | 0.657 | 0.614 | 0.403 | **0.528** |
| 2006 | | | X | 0.665 | 0.620 | 0.420 | **0.515** |
| 2007 | | | | X | 0.671 | 0.493 | **0.566** |
| 2008 | | | | | X | 0.512 | **0.590** |

The observed lack of correlation could be a substantive product of load shedding on the Syrian electrical grid or an artifact of a shift in the position of the digital numbers which occurred in 2009 from previous F16 satellite composites of Syria. To evaluate the latter possibility, we examine the location of luminous areas along the same latitude. Panels (a) and (b) in Figure 1 highlight five specific latitudes (in green) that are well-situated to show changes in the position of well-lit urban areas, which shifted in DN position to the south and east from 2007 to 2009. Comparing the two images shows consistent shifts in the position of several major Syrian cities, from Latakia, to Homs, and Hama. The difference in light emission patterns across the two maps demonstrates a distinct pattern of nighttime light loss to the north of urban areas and gain to the south of urban areas, further suggesting a shift.

Because shifts cannot be dealt with through intercalibration techniques, we develop our own manual technique. First, we code control points across the same visually identifiable intersection of four pixels—the border between a bright pixel cluster and a relatively darker space—across both 2007 and 2009 maps. Our process of placing control points is illustrated in Figure 2, using the city of al-Qamishli.

Table 2 shows the locations of each control point, the steps taken to calculate the difference between years for each control point and the aggregate difference across all points. We chose points along all of the country's populated peripheral areas and its geographic center (falling in 11 of the 13 total governorates) to ensure that the shift was systematically similar across the country. Results show the average shift in pixel location to be 0.016 decimal degrees to the south and 0.004 decimal degrees to the east, and that this shift is significantly different from zero.

**Table 2.** Locations of control points used to evaluate shift from 2007 to 2009.

| | Longitude | | Latitude | | Difference | |
| --- | --- | --- | --- | --- | --- | --- |
| **Place** | **2007** | **2009** | **2007** | **2009** | **Long.** | **Lat.** |
| Homs | 36.721 | 36.715 | 34.845 | 34.835 | 0.006 | 0.010 |
| Hama | 36.744 | 36.740 | 35.186 | 35.173 | 0.004 | 0.013 |
| Aleppo | 36.970 | 36.965 | 36.393 | 36.373 | 0.005 | 0.020 |
| Jabla | 35.903 | 35.900 | 35.356 | 35.336 | 0.003 | 0.020 |
| al-Raqqa | 38.980 | 38.975 | 35.920 | 35.908 | 0.005 | 0.012 |
| Dayr al-Zur | 40.171 | 40.169 | 35.320 | 35.308 | 0.003 | 0.012 |
| Rural Damascus 1 | 36.588 | 36.584 | 33.503 | 33.486 | 0.004 | 0.018 |
| Rural Damascus 2 | 36.302 | 36.298 | 33.646 | 33.629 | 0.004 | 0.017 |
| al-Tal | 36.189 | 36.184 | 34.028 | 34.014 | 0.005 | 0.014 |
| al-Hasaka | 40.723 | 40.718 | 36.546 | 36.532 | 0.004 | 0.015 |
| al-Qamishli | 41.208 | 41.202 | 37.023 | 37.008 | 0.006 | 0.014 |
| Tartus | 35.910 | 35.906 | 34.862 | 34.839 | 0.004 | 0.023 |
| | | | | **Mean** | 0.004 | 0.016 |
| | | | | **Standard Deviation** | (0.001) | (0.004) |
| | | | | **T-value** | 15.112 | 13.761 |

*Note:* The longitude and latitude differences return a −0.31 correlation. The moderate correlation suggests a consistent direction to the underlying shift in visually identified changes in pixel position.

To correct for the shift across the entire 2009 composite, we utilize the "shift" command in the raster package in R. This moves the 2009 raster based on the mean observed shift in Table 2. The command, a single line of code that incorporates the calculated shift from Table 2, is shown below:

```
1  nightlights2009shift<−raster::shift(nightlights2009,dx=0.0043894,dy=0.015704)
```

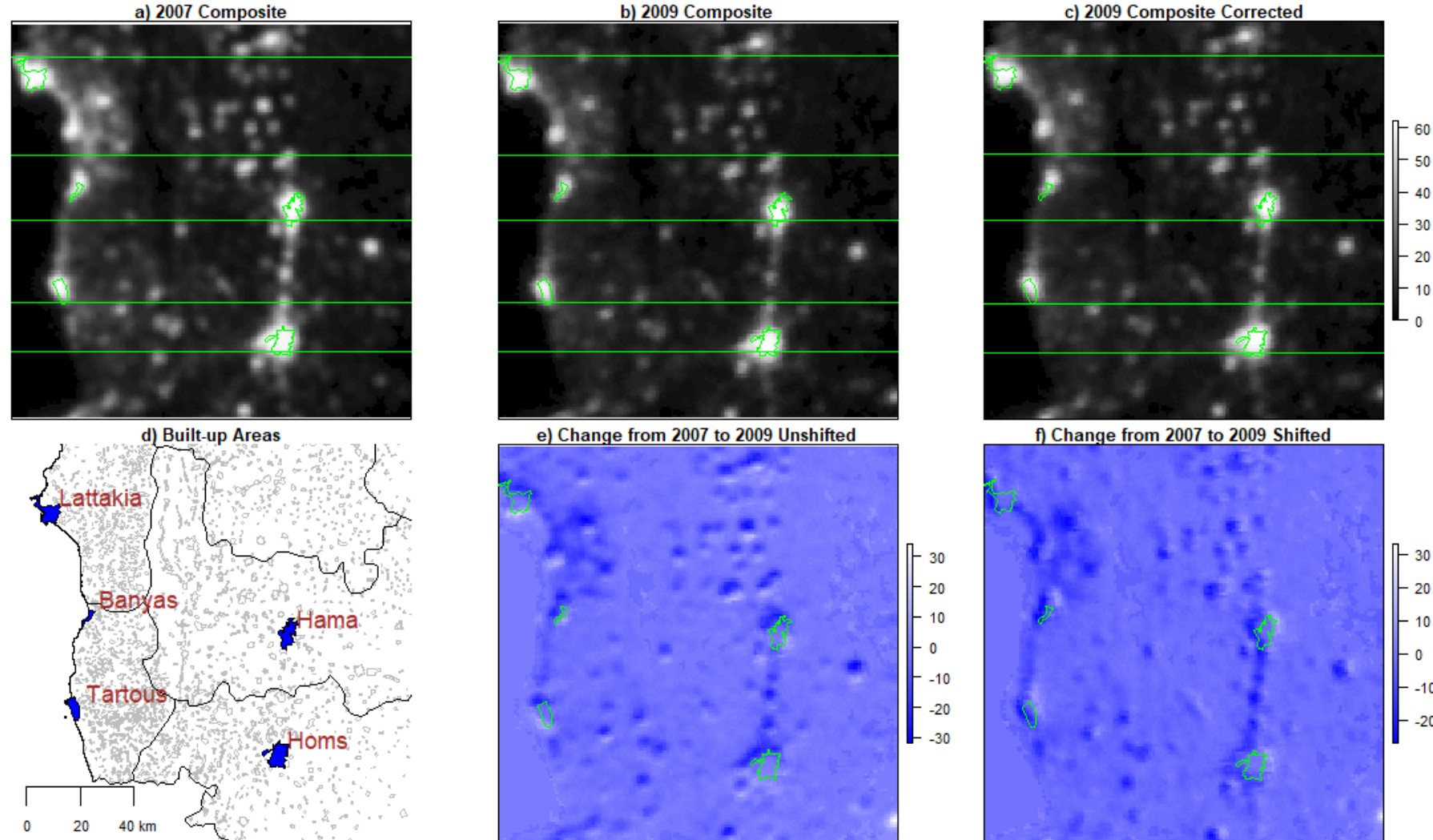

**Figure 1.** Evaluating signal shift in 2007 and 2009 F16 composites. Data on Syrian built-up areas obtained from the United Nations Office for the Coordination of Humanitarian Affairs [40].

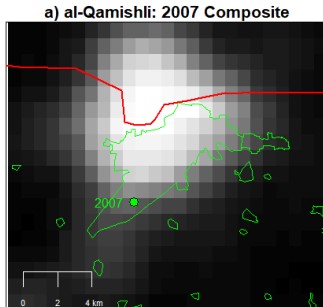 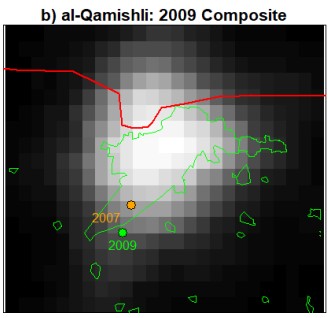 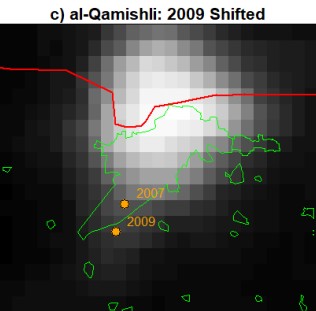

**Figure 2.** Control points outside of al-Qamishli (border with Turkey in red, built-up area in green).

This operation results in a noticeable synchronization between the images, as can be observed in panels (c) and (f) in Figure 1. Moreover, the systematic appearance of shifts is no longer evident in the corrected difference measure; Table 1 shows a notable improvement in correlation of the 2009 shifted composite with most of the remaining composites from the F16 satellite, in comparison to their correlation with the original 2009 composite.

In Figures S1–S4, we replicate our visual inspection for the remaining F16 coverage years and 2009. We observe a similar disjuncture between the 2009 composite and those for 2005, 2006, and 2008 to what we observe between the 2009 and 2007 composites. Visual inspection reveals no shift between 2004 and 2009, potentially explaining why the shifted 2009 map is less correlated with the 2004 composite than the original and suggesting that a similar shift may also be present in the 2004 composite.

## 3. Results

### 3.1. Does Signal Shift Affect Estimates of the Relationship between Load Shedding and Violence?

De Juan and Bank [31] examine the relationship between changes in light emission and large-scale violence in Syria's third-level administrative regions, known as sub-districts (*nawahi*, *n = 266*). The median area of a sub-district is 263 sq. km, with areas ranging from 6 sq. km (al-Hajar al-Aswad, a Damascus suburb) to 7105 sq. km (al-Dumayr, in the Syrian desert east of Damascus). The outcome measure of interest is a dichotomous variable recording whether a given sub-district experienced 25 or more deaths due to violence between the start of the uprising and November 2012, derived from Syria Tracker, an initiative to catalogue deaths in the early stages of the Syrian conflict. The main independent variable of interest is load shedding, measured as sub-district-level changes in average nighttime light emissions between 2007 and 2009. Several other demographic control variables, including population size, the ethnic identity of sub-district residents and the proportion of children enrolled in school, are also included. The study finds a significant negative relationship between increases in nighttime light emissions and the likelihood of violence and concludes that regime favoritism reduced the likelihood of violence at the onset of conflict.

In Table 3, we compare the results from the study's main analysis, which uses an uncorrected average difference in nighttime light intensity between 2007 and 2009, with results from a model using maps manually corrected for 2009. Our baseline model replicates the ideal models described [31], using unconditional governorate fixed effects and clustering standard errors at the governorate level. Clustering standard errors accounts for spatial autocorrelation, which is likely to be present in smaller sub-districts. We reproduce models using conditional fixed effects without clustered standard errors in Table S8. The difference between these specifications is negligible.

Model 1 in Table 3 replicates the original finding of a significant negative relationship between changes in nighttime light intensity and violence. After we apply a correction for signal shift, in model 2, the negative relationship becomes less pronounced and is no longer statistically significant. Models 3 through 5 add further adjustments for saturation and the presence of gas flares. Whereas the original study did not account for saturation and accounted for gas flares by removing all observations from 5 of the 13 governorates (see Figures S6–S8), we utilize data from Elvidge et al. [22] to remove only

10 kilometer buffers around gas flare positions. Tables S6 and S7 present further statistical models correcting for gas flares and using radiance calibrated images to account for saturation.

**Table 3.** Logistic regression of violence from March 2011 to November 2012 in Syria.

| Dependent Variable: | 25 or More Deaths in Sub-District (Binary) | | | | |
| --- | --- | --- | --- | --- | --- |
| | (1) | (2) | (3) | (4) | (5) |
| Gov. Employees | −0.012 | −0.014 | −0.008 | −0.012 | −0.006 |
| | (0.028) | (0.025) | (0.027) | (0.026) | (0.028) |
| Sunni | 0.369 | 0.439 | 0.365 | 0.414 | 0.353 |
| | (0.583) | (0.572) | (0.577) | (0.567) | (0.576) |
| Alawi | −0.022 | −0.035 | −0.066 | −0.011 | −0.044 |
| | (0.547) | (0.508) | (0.476) | (0.457) | (0.438) |
| School Enrollment | −0.046 | −0.029 | −0.006 | −0.012 | 0.009 |
| | (0.113) | (0.116) | (0.136) | (0.111) | (0.127) |
| Border Dist. (log) | −0.003 | −0.045 | −0.119 | −0.091 | −0.166 |
| | (0.471) | (0.488) | (0.535) | (0.465) | (0.510) |
| Urbanization | 2.932 *** | 2.672 *** | 2.766 *** | 2.583 *** | 2.705 *** |
| | (0.660) | (0.524) | (0.555) | (0.559) | (0.587) |
| Electrification | −0.008 | 0.007 | 0.000 | 0.007 | −0.001 |
| | (0.100) | (0.094) | (0.119) | (0.100) | (0.125) |
| Pct. Unemployed | −0.009 | −0.005 | −0.007 | −0.003 | −0.005 |
| | (0.029) | (0.027) | (0.026) | (0.025) | (0.024) |
| Road Density | −2.590 *** | −1.919 *** | −2.838 *** | −1.964 *** | −2.769 *** |
| | (0.578) | (0.446) | (1.031) | (0.448) | (1.024) |
| Population (log) | 1.932 *** | 1.826 *** | 1.981 *** | 1.871 *** | 2.032 *** |
| | (0.701) | (0.634) | (0.495) | (0.630) | (0.513) |
| Lights Change (Original) | −0.464 *** | | | | |
| | (0.121) | | | | |
| Lights Change (Shifted) | | −0.359 | | | |
| | | (0.225) | | | |
| Lights Change (Shifted, No Saturation) | | | −0.264 | | |
| | | | (0.246) | | |
| Lights Change (Shifted, No Flares) | | | | −0.250 | |
| | | | | (0.187) | |
| Lights Change (Shifted, No Sat. No Flares) | | | | | −0.165 |
| | | | | | (0.190) |
| Constant | −1.791 | −4.060 | −6.986 | −6.194 | −8.781 |
| | (17.823) | (14.744) | (14.760) | (16.400) | (16.307) |
| Observations | 247 | 247 | 247 | 247 | 247 |

*Note:* \*$p < 0.1$; \*\* $p < 0.05$; \*\*\* $p < 0.01$. All models include first-level administrative (governorate) fixed effects and governorate-clustered standard errors. Model 1 replicates model 2 on Table 1 in De Juan and Bank's (2015) article.

Jointly correcting for signal shift, saturation and gas flares in models 3 through 5 moves the coefficient of nighttime light changes from 2007 to 2009 further toward zero. In summary, when we correct for signal shift, the relationship between changes in mean nighttime light intensity and violence in Syria loses substantive and statistical significance.

*3.2. Did Signal Shift Produce Systematic Bias?*

Because we could not replicate the previously significant finding when correcting for signal shift, a question arises regarding whether signal shift introduces systematic bias. Both the placement of some city lights in the Mediterranean Sea (Figure 1b) and the spillover of lights from the Turkish town of Nusaybin (Figure 2b) suggest systematic change due to signal shift. To pursue this issue further, we analyze the relationship between the demographic variables in Table 3 and the difference between the original F16 2009 composite and the shifted map. The formula for the outcome variable is as follows:

$$\Delta NL_{2009} = (NL_{2009} - NL_{2007}) - (NLShift_{2009} - NL_{2007})$$
$$\Delta NL_{2009} = NL_{2009} - NLShift_{2009} \tag{1}$$

The outcome variable is positive if the unshifted map overstated the mean luminosity of a region in 2009 and negative if the unshifted map understated the 2009 mean. In other words, sub-districts

with positive values experienced more load shedding than the original maps indicate, while those with negative values experienced less. Figure 3 shows the original average DN changes, the average changes with 2009 shifted, the difference between the two and the original dependent variable.

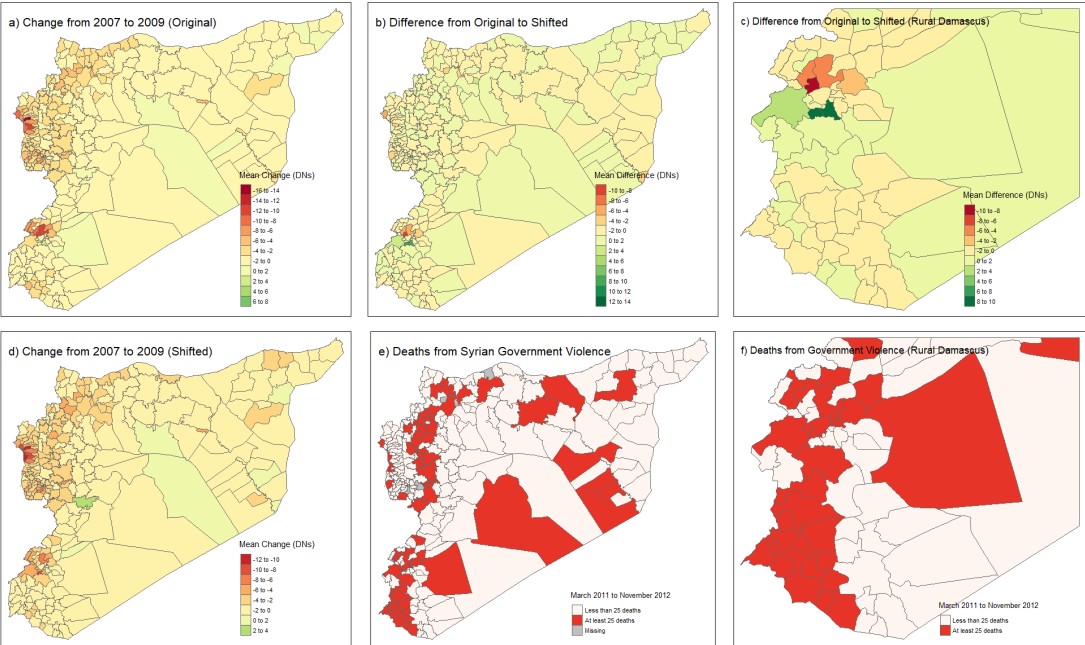

**Figure 3.** Changes in digital number average across Syrian sub-districts.

The area around Damascus (Figure 3c) shows the starkest differences from the original to the shifted map. Sub-districts to the southwest of Damascus, in the rural Damascus governorate, experience the highest gains in DNs and sub-districts to the north of Damascus and the city center itself experience the highest losses. With the north of the Damascus area experiencing more violence than the southwest, the pattern suggests less of an association between DN change and violence than previously observed.

Moving to a multivariate specification, we utilize the demographic covariates of Table 3 but change the dependent variable to the difference between the 2009 unshifted and shifted composites. Due to the continuous nature of our outcome variable, we utilize an ordinary least squares regression, adding governorate-level fixed effects and clustering standard errors at the governorate level.

Table 4 presents our findings. The second model is of particular interest because it excludes changes in areas in which there were gas flares; systematic shifts in gas flares are unrelated to load shedding and tend to be in unpopulated areas of sub-districts. In model 2, we find that more populated sub-districts were more likely to have more pronounced declines in mean DNs for 2009 when adjusting for signal shift; in other words, correcting for signal shift allows us to observe more negative changes in nighttime lights from 2007 to 2009 in more populous areas. This suggests that failing to account for signal shift created systematic measurement error, understating load shedding in more populous areas and giving the appearance that locations in which nighttime light averages declined were more likely to experience violence. While apparently random, signal shift can introduce systematic bias into differences between annual DMSP-OLS composites by undercounting changes in nighttime light intensity in more populous areas.

**Table 4.** Ordinary least squares (OLS) regression to assess systematic error.

| Dependent Variable: | Δ Mean Sub-District DNs after Adjusting for Signal Shift | | | |
|---|---|---|---|---|
| | **(1)** | | **(2)** | |
| | **Coef.** | **S.E.** | **Coef.** | **S.E.** |
| Gov. Employees | −0.002 | (0.004) | −0.005 | (0.006) |
| Sunni | −0.185 | (0.211) | −0.438 | (0.364) |
| Alawi | 0.020 | (0.195) | −0.265 | (0.255) |
| School Enrollment | 0.031 *** | (0.012) | 0.032 | (0.044) |
| Border Dist. (log) | 0.047 | (0.082) | 0.134 | (0.182) |
| Urbanization | 0.467 | (0.387) | −0.752 * | (0.432) |
| Electrification | −0.008 | (0.013) | 0.066 | (0.065) |
| Pct. Unemployed | 0.011 ** | (0.005) | 0.016 * | (0.009) |
| Road Density | −0.965 | (0.710) | −0.011 | (0.309) |
| Population (log) | 0.036 | (0.191) | 0.259 ** | (0.107) |
| Constant | −1.878 | (1.499) | −13.164 *** | (3.339) |
| R-squared | 0.050 | | 0.067 | |
| Gas Flares | Included | | Excluded | |
| Observations | 247 | | 247 | |

Note: * $p < 0.1$; ** $p < 0.05$; *** $p < 0.01$. All models include first-level administrative (governorate) fixed effects and governorate-clustered. standard errors.

## 4. Conclusions

We identify a compatibility issue in some DMSP-OLS composites from the same satellite: the geoposition of digital numbers for the same physical location can shift from one year to another. We employ moving window raster correlations to identify years in which shifts may have taken place, visually identify the shift, and employ control points to synchronize previously shifted annual composites. We then use the shifted composites to re-examine a recent study by De Juan and Bank [31] that utilized the differences across two DMSP-OLS composites to predict violence during the Syrian uprising. In our re-analysis, we observe the possibility for false positives in statistical results utilizing shifted composites, which may be a result of systematic bias introduced by signal shift.

Whether researchers using DMSP-OLS data will need to make corrections for signal shift depends on how they utilize the data. Scholars examining the differences between individual DMSP-OLS composites (e.g., [4,31]) need to be most concerned with issues of signal shift; they must either plausibly rule out signal shift or adjust for it. Scholars utilizing a single annual composite of DMSP-OLS data need to be least concerned about this effect (e.g., [19]), though composites with signal shift may be more subject to measurement error when paired with other geolocated data. Finally, analyses utilizing DMSP-OLS composites in cross-sectional time-series (TSCS) models (e.g., [18]) are not necessarily subject to the same problems as those using the differences between two composites, provided they utilize year fixed effects to account for unique year characteristics in DMSP-OLS data, including signal shift.

Our findings also have implications for scholars of Syria, suggesting that factors other than electricity distribution played a role in impelling violence. We contextualize these findings in Figure S7–S9, qualitatively investigating the links between nighttime lights, patronage and violent contention through local news sources. These materials suggest that load shedding may not be tightly linked to changes in observed nighttime light intensity and that the Syrian energy sector was not a vehicle for the sorts of clientelism relevant to contentious challenge.

Finally, we recognize that an automated or semi-automated process would offer a more efficient way of both detecting and correcting for signal shift. We envision our technique being a building block for such a process to be developed and for its results to be validated. Visual inspection is integral to the construction of a model for detecting signal shift systematically, and our control point methods can be used to evaluate the effectiveness of automated processes once they are available. Broadly, we hope that our study will encourage other researchers to undertake additional efforts to deal with signal shift.

**Supplementary Materials:** The following are available online at http://www.mdpi.com/2072-4292/12/14/2219/s1, Table S1: F10 Satellite Interannual Pearson Correlations, Table S2: F12 Satellite Interannual Pearson Correlations, Table S3: F14 Satellite Interannual Pearson Correlations, Table S4: F15 Satellite Interannual Pearson Correlations, Table S5: F18 Satellite Interannual Pearson Correlations, Table S6: Logistic Regression on Violence in Syria Adjusting for Saturation and Gas Flares, Table S7: Logistic Regression on Violence in Syria with Radiance-Calibrated DMSP-OLS Data, Table S8: Logistic Regression on Violence in Syria Applying Corrective Shift - Conditional Fixed Effects, No Clustered Standard Errors, Figure S1: Evaluating Signal Shift in 2004 and 2009 F16 Composites, Figure S2: Evaluating Signal Shift in 2005 and 2009 F16 Composites, Figure S3: Evaluating Signal Shift in 2006 and 2009 F16 Composites, Figure S4: Evaluating Signal Shift in 2008 and 2009 F16 Composites, Figure S5: Change from original to intercalibrated differences from 2007 to 2009 DMSP-OLS data, Figure S6: Gas Flaring in Eastern Syria (latitude 34-36, longitude 38-40), Figure S7: Overlay of areas excluded by De Juan and Bank (2015) due to gas flares (blue) and gas flares shapefile from NOAA (red), Figure S8: DMSP-OLS F16 2007 Composite with 63 DN pixels in red, Figure S9: Percentage change in nightlights, power consumption, and peak demand in Syria. Replication code for all statistical analysis performed in the article.

**Author Contributions:** Conceptualization, K.A. & K.M.; methodology, K.A. & K.M.; software, K.A. & K.M.; investigation, K.A. & K.M.; data curation, K.A. & K.M.; original draft preparation, K.A. & K.M.; review and editing, K.A. & K.M.; visualization, K.A. All authors have read and agreed to the published version of the manuscript.

**Funding:** This research received no external funding.

**Acknowledgments:** We thank Chris Barrie, Nils Metternich, Raphaël Lefèvre, Matthew Nanes and Nils Weidmann for their helpful feedback.

**Conflicts of Interest:** The authors declare no conflict of interest.

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
