# Peer review of "Identifying and Correcting Signal Shift in DMSP-OLS Data"

_remotesensing, doi:10.3390/rs12142219_

Round 1

Reviewer 1 Report

The manuscript reports experimental realization of shifts in digital number position based on the use of the DMSP-OLS nighttime light data. Detailed and systematic experiments based on the DMSP-OLS nighttime light data are presented, providing a thorough characterization of the shifts in digital number position. The results reported represent a notable advance in the improvement of the DMSP-OLS nighttime light data.

In my opinion, the manuscript is suitable for publication in Remote Sensing, after the authors have addressed the following comments.

1) Page 1, line 1. You mentioned “DMSP-OLS nighttime light data has become a mainstay in social science”. Make the sentence as environmental and socio-economic fields.

2) Page 1, line 15. You mentioned “Since then, DMSP-OLS nighttime lights data has been made available for each year between 1992 and 2014”. The data are available from 1992 to 2013. Thus, you have a choice to say between 1992 to 2013 OR between 1991 and 2014.

3) Page 1, lines 16-17. Add some references for environmental studies.

4) Page 1, lines 18-19. You mentioned “DMSP-OLS data cannot capture variation in high density urban areas due to pixel saturation”. I disagree because there is a variety of attempts have been made to address the saturation effect by using NDVI, and land use/cover data. Have a look to this paper

Urban mapping using DMSP/OLS stable night-time light: a review

Published in International Journal of Remote Sensing (2017)

Authors: Li and Zhou

5) Page 1, line 25. You mentioned “Our note identifies an additional limitation that has not been explored in previous studies”. I believe it will be better to replace a word of “explored” with “examined”.

6) Page 2, line 33. You mentioned “For the remainder of the note, we outline how to identify signal shift across two different annual DMSP-OLS maps, how to resolve the issue and how to test for whether findings from past studies have been affected by signal shift”. Only one study (DJB) is tested not many studies.

I suggest re-writing the sentence and saying something like:

For the remainder of the note, we outline how to identify signal shift across two different annual DMSP-OLS maps, how to resolve the issue and how this problem can affect findings.

7) Page 2, line 35. You mentioned “Notably, our strategies for addressing signal shift are distinct from strategies to correct for signal decay”. Distinct from which strategies.

8) Page 2, line 63. You mentioned “We correct for the shift and find that DJB’s findings are no longer statistically significant”. Say something like “We correct for the shift and find that DJB’s findings can be improved”

In academic research, it is preferable to indicate that the results of X work can be improved rather than indicating the results are wrong. Try to concentrate on how the signal shift affects findings without mention DJB. Please do this for the whole manuscript.

9) Page 2, lines 64-65. You mentioned “We close our note with suggestions to researchers using DMSP-OLS data in future studies”. There is no meaning of this sentence. Re-write

10) Page 3, Figure 1. What is the source of built-up areas?

11) Page 3, Figure 1. The city names are overlapped with the built-up areas.

12) Page 4, line 93. The name must be changed.

13) Page 5, Table 2. The name of the table must be changed.

14) Page 6, line 121. You mentioned “The findings cumulatively suggest that failing to adjust for signal shift led to a spurious finding in DJB’s article”. Say something like

The findings cumulatively suggest that failing to adjust for signal shift will affect the result and thereby affect the conclusion.

15) Page 8, line 158. You mentioned that “Scholars that most need to be concerned are those like DJB [e.g., 4]”. Re-write.

16) Page 9, line 231 (Ref. 21). The reference was wrongly written.

If I were you, I will do the experiment without mentioning DJB. I will write a simple sentence in the discussion part and say “The results of this work are different from those obtained by DJM (This is not a comment it is an advice).

Reviewer 2 Report

The authors build on results reported by Tuttle et al in 2013, who showed a 2.9 km error in the geolocation of DMSP OLS imagery. This paper refers to the archive of annual composites of OLS imagery, and claims to identify systematic shifts in the indicated position of fixed lights in Syria between 2007 and 2009. The aim of doing so is to demonstrate that a previously published analysis that draws inferences from changes in night lights in Syria over the same period is invalidated because these shifts were not identified or corrected for.

I am reasonably confident that the authors of the present manuscript have correctly identified a change in spatial registration between the two image composites that they have analysed, although their method of doing so is (as they admit) somewhat crude. What remains of concern is the fact that they do not clearly enough explain how they choose their control points, nor discuss the fact that the shift between dates is smaller than the spatial resolution of the image data. Their observations that spatial errors have the ability to confound inferences drawn from geographical units (the third-level administrative divisions) that are not large compared with the size of the errors is undoubtedly correct. However, I would like to see some discussion of the match between the spatial scale of the Nawahi in the most densely populated areas and the spatial resolution of the OLS data – in other words, whether there was any real likelihood that the DJB analysis could have produced meaningful results even if the spatial shift had been corrected.

32. The years 2007 and 2009 were chosen for the present study because these were also used in the work of De Juan and Blank (DJB) and the authors here wish particularly to show that the conclusions of the DJB study were incorrectly drawn from the data. This is valid, but the generality of the results presented here would be increased by analysing other years too. Would it be hard to add a few more? Also, the authors don’t say whether they used the F15 or F16 data from 2007.

67. Isn’t the first step to obtain the OLS images? Where were they obtained from? Please give details. I downloaded the dataset F162009 (the only one calculated for 2009, as far as I am aware) from https://eogdata.mines.edu/dmsp/downloadV4composites.html

to check the coordinates.

74. ‘suggesting the presence of a skew as a result of signal decay’. I do not think there is any evidence for a skew in the data, in the geometrical sense, just a shift. And I don’t think the authors have presented any evidence for the shift being due to signal decay.

Fig. 1 What are the red lines and the grey contours? What is the scale of these images?

77 Manual technique. Although the consistency of the results in table 1 do suggest that the manual method is satisfactory, I have some reservations about it and would like to know more details about it before I can agree that it is sufficiently rigorous. (1) It isn’t clear how the control points’ specific pixels, which have to do duty as spatial invariants, have been selected. For example, I examined the pixel in the 2009 image corresponding to the Tartous control point (last in table 1). Assuming I hit the right pixel, it has a DN of 22, which is not higher than all of its eight immediate neighbours, so since it is not a local maximum I am not clear how you could be sure to pick the same pixel in the image from another year. (2) Since the pixels in the image are 30 arcseconds ≈ 0.008 degrees in size, it is not immediately obvious that the coordinates of a pixel can be specified by manual interrogation to a precision of 0.001 degrees. That would require being able to differentiate between around 70 different positions within one pixel. (3) The actual spatial resolution of the OLS radiometer is at best (near nadir) 2.2 km, i.e. around 0.02 degrees, so an area of roughly 4 x 4 pixels is strongly correlated. In the light of these concerns, I would be much more convinced by a method of detecting the shift between image dates that used spatial cross-correlation or similar.

Table 1. Suggest replacing ‘X coordinate’ and ‘Y coordinate’ with ‘longitude’ and ‘latitude’ respectively, and replacing ‘aggregate’ with ‘mean’. The table could be more economically laid out, with just one line per place and still seven columns per line (place, 2007 longitude, 2009 longitude, 2007 latitude, 2009 latitude, long difference, lat difference). Would be good also to add the standard deviations of these differences, to be sure that they are in fact significant. Also it would be good to establish that the shifts are not spatially correlated (e.g. shifts predominantly in one direction in one part of Syria, another in a different part. My quick check, just by correlating shifts against latitude and longitude, was reasssuring. But it would be good to hear from the authors on this point.) By my calculations, the shifts are 0.0044 (0.0012) in longitude, and 0.0160 (0.0043) in latitude, both of which are significantly different from zero (p<0.005 for both).

Table 2. Lacks enough detail to be understood without having the DJB paper in front of me. What are the numbers in the table?

110 ‘their attempt to account for gas flares falls short of being a reliable technique.’ Please elaborate on this remark in the text.

122-123. Equation (1) needs an ‘=’.

Reviewer 3 Report

Review comments: Manuscript ID:- remotesensing-853525

General Comment

This is an interesting manuscript about addressing shifts of Defense Meteorological Satellite Program Operational Linescan System data. However, there are aspects that require improvement and clear presentation before being considering for publication. Specific comments and suggestions are included below.

Specific comments and suggestions

L1: First define the abbreviation.

L8: Authors need to brief the kind of their suggestion as conclusion in the abstract part.

L18: Which problem/limitation mentioned earlier to led to mention in the sentence “…DMSP-OLS nighttime lights data also has several important limitations?”

L19: What about the effect of pixel resolution?

L23: I suggest authors for adding sources that studies are still using the DMSP-OLS data.

L25: “..note..” or study?

L26-27: Authors need to include the source of such systematic shift, if already elsewhere in literature or to mentioned it as part of their study.

L34: Distinct in what way?

L46-48: What if the signal shifts from one year to other year systematic, will not be possible the shift (error) of the two-time window offset each other?

L67: How to avoid such kind of labor-intensive type of identification through inspection? Are there any suggestion to make a fully (semi)-automatic identification?

L66-177: In the current version of the manuscript, there is mix-up of sections. I suggest following the journal guideline, such as, materials and methods, results and Discussions (which the discussion section can also be independent) – in both cases – the finding should be properly discussed, which is not the case in the current version. Finally the concussions section, to conclude the major findings and suggest future area of study if needed.

Round 2

Reviewer 2 Report

The revised manuscript is much sharper than the original, and has addressed all the points that were of concern. I now really like this paper and would like to see it in published.

Reviewer 3 Report

Review comments: Manuscript ID:- remotesensing-853525

The manuscript (ID: remotesensing-853525) entitled with “Identifying and Correcting Signal Shift in DMSP-OLS Data” has gone through a significant revision as compared to the earlier version. Major issues from my side were already taken into account.

The manuscript has merit as an attempt of developing techniques for identifying and correcting signal shift of Defense Meteorological Satellite Program Operational Linescan System (DMSP-OLS) data. The study shows ways of identifying signal shift between two DMSP-OLS maps, and how such shift can affect findings. Thus, I recommend considering this manuscript for publication.